

# A case for myoglobin-macromolecular rate theory applied to pseudo peroxidase kinetics

Collin Tuttle[1], Michael Hannesson[1], Amy Henrichsen[2],
Lily Hainsworth[2], Camille Condie[2], Aj Whitesides[2], Archel Oren[2],
Simeon Tanner[1], Benjamin Terry[1], Jacob Cannon[1], Jeremy Johansen[1],
Alisha Bhatia[3] and Daniel Scott[1]

[1] Chemistry, Utah Valley University, Orem, Utah, United States
[2] Biochemistry, Brigham Young University Hawaii, Laie, Hawaii, United States
[3] Chemistry, University of California, Los Angeles, Los Angeles, California, United States

Corresponding author
Daniel Scott, daniel.scott@uvu.edu

## ABSTRACT

This study explores the well-known catalytic behavior of myoglobin as a pseudo-peroxidase by applying macromolecular rate theory (MMRT) to assess its temperature-dependent enzyme kinetics. While myoglobin is primarily recognized for its oxygen-binding properties in muscle tissues, with a characterized pseudo-peroxidase ability to catalyze the degradation of hydrogen peroxide in the presence of electron donors, the claim that myoglobin is actually a true peroxidase can be explored by analyzing the heat capacity changes ($\Delta C_p^{\ddagger}$) in the catalyzed reaction at different temperatures and fitting the results to the expanded Eyring equation (MMRT equation). This research uses the MMRT equation to compare myoglobin's catalytic activity (a pseudo-peroxidase) with that of lactoperoxidase (a true peroxidase) and copper ions (a non-enzymatic catalyst) across a range of temperatures at pH 5, after which the biological catalysts are compared again at pH 7. By analyzing the $\Delta C_p^{\ddagger}$ of these catalysts, it was found that myoglobin exhibits a significant catalytic contribution at both pH levels, suggesting a structural/vibrational or some other relatively significant transition during the reaction. The study's findings provide a new perspective into myoglobin's enzymatic role in peroxide decomposition and highlight the utility of MMRT in quantifying the contribution of polypeptide chains in enzyme-catalyzed peroxidase reactions. Additionally, our research notes the pH-dependence of myoglobin's catalytic efficiency compared to traditional peroxidases, offering implications for understanding its broader biological roles.

## INTRODUCTION

Myoglobin is an oxygen-binding protein found mainly in skeletal and heart muscles. It is known for its finely tuned oxygen-binding properties, which allow it to maintain the homeostatic concentration of molecular oxygen used for metabolic processes (*Wittenberg & Wittenberg, 2003*; *Polasek & Davis, 2001*). Myoglobin contains a covalently bound

iron-coordinating heme molecule where molecular oxygen-binding occurs. Although myoglobin is one of the earliest molecularly characterized proteins, it could be argued that the complete role of myoglobin is not fully understood (*Koch et al., 2016*). For example, mice that have the gene for myoglobin removed from their genome are able to survive with reproductive viability; albeit with compensatory mechanisms including increased capillary density, coronary flow, and hemoglobin levels (*Garry et al., 1998*). Additionally, myoglobin is also found to work in niche roles in different fringe "cycles" or processes (*Flözgel et al., 2001*; *Flögel et al., 2004*; *Hardison, 1996*; *Ordway & Garry, 2004*). Myoglobin is less known for its catalytic capabilities and can currently be categorized as a pseudo-peroxidase as opposed to a true peroxidase like lactoperoxidase (*Huo et al., 2021*). Lactoperoxidase and myoglobin are heme-containing proteins found in mammals, but they are not closely related evolutionarily (*Sharma, Kapoor & Gulati, 2013*). Lactoperoxidase is an enzyme involved in antimicrobial defense, found in milk and secretions, where it catalyzes the oxidation of substrates using hydrogen peroxide to inhibit microbial growth. Myoglobin, on the other hand, is a globular oxygen-binding protein found in muscle tissue, responsible for storing and facilitating oxygen transport to support muscle activity.

Structurally, lactoperoxidase belongs to the peroxidase-cyclooxygenase superfamily, while myoglobin is part of the globin family. The heme group in lactoperoxidase acts as a catalytic cofactor in oxidative reactions, whereas in myoglobin, it enables oxygen storage and release. These structural and functional differences suggest that the two proteins evolved from separate ancestral lineages. Lactoperoxidase shares evolutionary ties with other peroxidases, such as horseradish peroxidase, while myoglobin shares ancestry with hemoglobin and other globins specialized for gas transport. Although both proteins utilize the heme group, this is an example of convergent evolution, where unrelated proteins independently adopt similar molecular motifs for different purposes.

While the heme group itself is an ancient molecular feature, its integration into lactoperoxidase and myoglobin represents distinct evolutionary paths tailored to different physiological roles. Lactoperoxidase's evolution within the peroxidase family reflects its role in oxidative defense, while myoglobin's development within the globin family underscores its specialization in oxygen storage and delivery (*Endeward, Gros & Jürgens, 2010*). This functional divergence underscores how evolutionary processes repurpose shared biochemical features like the heme group to address different biological challenges.

As shown in Fig. 1, the structural differences between myoglobin and lactoperoxidase are significant (*Singh et al., 2009*), even though they do both contain the heme compound with the coordinated Fe ion. As mentioned, myoglobin can behave catalytically toward the degradation of hydrogen peroxide in the presence of an electron donor, but it is generally accepted that this is not necessarily a primary function of myoglobin but more of an accident, or what in this work will be referred to as an unintended reaction (*Carlsen, Skovgaard & Skibsted, 2003*; *Hersleth et al., 2007*; *Vlasova, 2018*).

Of the biologically significant reactive oxygen species (ROS), hydrogen peroxide is considered less reactive than species such as nitric oxide, hydroxyl radicals, or superoxide anions. Upon being produced in the mitochondrial electron chain, superoxide anions are converted directly to hydrogen peroxide *via* superoxide dismutase (*Lennicke et al., 2015*).

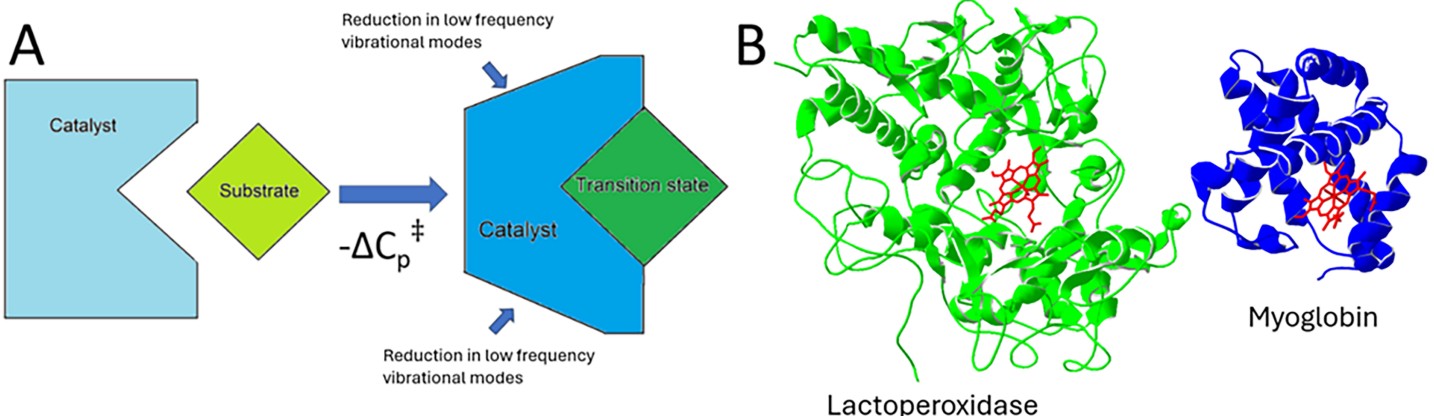

**Figure 1 Depiction of enzyme transition state changes and a comparison between myoglobin and lactoperoxidase.** (A) This illustration of a catalyst-transition state complex demonstrates a reduction in low frequency vibrational modes. The change in these vibrational modes results in a change in the catalyst heat capacity. (B) Ribbon models of bovine lactoperoxidase and myoglobin. Lactoperoxidase is a true peroxidase while myoglobin is a pseudo peroxidase. Both catalyze the peroxidase reaction *via* the heme group, shown here in red in each catalyst.

Due to its status as an ROS, hydrogen peroxide has mostly been studied for the deleterious effects caused by its reactive interactions with biological macromolecules. As the field of cellular signaling has grown in recent years, the role of hydrogen peroxide as a signaling molecule rather than as a harmful by product has been increasingly investigated (*Stone & Yang, 2006*). These mechanisms can include ones similar to the manipulation of reversible reactions in "redox-sensitive" protein residues and observed activating/mediating effects in a variety of enzymatic systems (*Jackson, Stretton & McArdle, 2020*; *Lennicke et al., 2015*). Given the developing body of knowledge regarding the importance of peroxides in biochemical systems, it would be reasonable to say that proteins capable of scavenging peroxide molecules (such as myoglobin) would be duly affected by selective mechanisms toward this capacity.

Biological peroxidases and pseudo-peroxidases are recognized as generally being heme-bearing catalysts (typically protoporphyrin IX or a similar derivative) that catalyze the degradation of reactive oxygen species such as peroxides. Pseudo-peroxidases differ from peroxidases in a variety of ways and the exclusion of myoglobin from the peroxidase categorization is completely understandable (*Vlasova, 2018*); however, the delineation between these two groups could be better informed with quantitative analysis. Figure 1 illustrates the ribbon models of bovine lactoperoxidase and myoglobin, demonstrating their structural differences and the heme group (in red) involved in catalysis.

For example, one of the criteria of a peroxidase is the presence of a distal amino acid ligand that can contribute, even if in a manner that is not fully understood, to the reaction's catalysis. The presence of such ligands would suggest a catalytic contribution of the polypeptide chain as opposed to just an exposed heme that is more indicative of an unintended reaction.

Arguably, another more general distinguishing characteristic of a catalyst that is enzymatic and reaction-specific in nature would be a significant positional/vibrational shift
in the polypeptide chain in the transition state formation (*Arcus & Pudney, 2015*). This could be manifested by, for example, a concerted movement of the chain or protonation or deprotonation of residues during catalysis, both of which are examples of contributions that have been suggested to be related to enzymatic catalysis.

In recent decades, a tool has emerged that may be able to quantify the contribution of the catalyst's polypeptide chain to a catalytic reaction. This tool emerged from questions regarding the dependence of reaction rates, specifically biological enzymatic reactions, on temperature. Reaction rate dependence on temperature has been a well-studied field of chemistry for decades, producing many useful kinetics models and equations that accurately describe the behaviors of reaction rates, including the rate trends of enzyme-catalyzed reactions.

Around 2010, research began to emerge suggesting that a lack of adherence of enzymatic data to the Arrhenius or Eyring equations (Eqs. (1) and (2)) was possibly not best attributed to protein denaturation or inactivation (*Daniel & Danson, 2010*).

$$k = Ae^{\left(\frac{-E_A}{RT}\right)} \tag{1}$$

$$k = \frac{k_B T}{h} e^{\left(-\frac{\Delta G^{\ddagger}}{RT}\right)}. \tag{2}$$

Specifically, the decrease in enzyme activity at higher temperatures has generally been attributed to catalyst denaturing or inactivation, as it is seen as a deviation from Eqs. (1) and (2). However, from an expansion of the Eyring equation to delineate not just the enthalpic ($\Delta H^{\ddagger}$) and entropic ($\Delta S^{\ddagger}$) contribution but also that of the heat capacity's ($\Delta C_P^{\ddagger}$) impact on the change in transition state energy (Eqs. (3) and (4)), macromolecular catalytic behavior at higher temperatures can be well explained (*Hobbs et al., 2013*; *Arcus et al., 2016*).

$$k = \frac{k_B T}{h} e^{\left[\frac{-\Delta H^{\ddagger} - \Delta C_P^{\ddagger}(T - T_0)}{RT} + \frac{\Delta S_{T_0}^{\ddagger} + \Delta C_P^{\ddagger}(\ln T - \ln T_0)}{R}\right]} \tag{3}$$

$$\ln k = \ln \frac{k_B T}{h} - \left[\frac{\Delta H^{\ddagger} + \Delta C_P^{\ddagger}(T - T_0)}{RT}\right] + \left[\frac{\Delta S_{T_0}^{\ddagger} + \Delta C_P^{\ddagger}(\ln T - \ln T_0)}{R}\right]. \tag{4}$$

This macromolecular rate theory (MMRT) details the dependence of a plot of ln k *vs.* temp (Eq. (4)). In a reaction where $\Delta C_P^{\ddagger}$ for a catalyst is essentially zero, Eqs. (3) and (4) collapse to Eq.(2) and Eyring kinetics are expected. However, when $\Delta C_P^{\ddagger}$ is, for example $-1.0$ kJ mol$^{-1}$ K$^{-1}$, there is a marked change in the shape of a curve that relates the ln k to temperature.

In short, it is suggested that the $\Delta C_P^{\ddagger}$ contributed to the observed deviations from Eq. (2) at higher temperatures (*Hobbs et al., 2013*; *Arcus et al., 2016*). Additionally, fitting plots of the ln k at different temperatures allows for the identification of $\Delta C_P^{\ddagger}$. Because this $\Delta C_P^{\ddagger}$ can

potentially be considered an indirect measurement of the catalysts' overall change as evidenced by a different capacity to absorb heat, it may also serve as an indicator or differentiator as to whether or to what degree the catalysts' contribution comes from a small catalytic change or an overall larger number of changes in bond rigidity or structure changes in the catalyst. If the catalysis of a reaction is mostly unintended, then the amino acid sidechain would not be evolutionarily chosen to contribute to catalysis, and a $\Delta C_p^{\ddagger}$ closer to zero would be expected. However, if there is a significant $\Delta C_p^{\ddagger}$, then it could be that a large portion of the catalyst contributes to the catalysis.

This results in a tool that may be useful to gauge the degree of contribution of the catalyst in a given chemical reaction, or at least the $\Delta C_p^{\ddagger}$ as enzyme and substrate transition to an enzyme-intermediate complex. Applying this tool to the case of myoglobin as a catalyst could further give insight as to its "role" as not only an oxygen-binding protein, but to what degree it may be behaving like a scavenging peroxidase, perhaps simply because of its abundance in organisms. As MMRT calculations result in a numerical value for $\Delta C_p^{\ddagger}$, these measurements provide a quantitative metric that can be insightful to the contribution of catalytic proteins.

There has been a lot of discussion from many perspectives on what contributes to non-Arrhenius behavior of biocatalysts. From the perspective of many, a large, negative $\Delta C_p^{\ddagger}$ during catalysis suggests tight binding at the transition state and a significant reduction in conformational flexibility, which is consistent with principles of statistical thermodynamics (*Feller & Gerday, 2003*; *Arcus et al., 2016*). Transition state theories also attribute this phenomenon to decreased vibrational and rotational modes at the activation barrier, emphasizing the influence of protein dynamics on catalysis (*Henzler-Wildman et al., 2007*). These perspectives highlight how $\Delta C_p^{\ddagger}$ may not only explain deviations from the Arrhenius equation but also provide insights into the molecular mechanisms underpinning enzymatic efficiency (*Hobbs et al., 2013*).

The magnitude of $\Delta C_p^{\ddagger}$ correlates strongly with enzyme rigidity, explaining the thermal adaptations observed in thermophilic and psychrophilic enzymes. For instance, thermophilic enzymes exhibit higher $\Delta C_p^{\ddagger}$ values, consistent with increased structural rigidity to withstand elevated temperatures. Conversely, psychrophilic enzymes demonstrate reduced $\Delta C_p^{\ddagger}$, favoring flexibility at lower temperatures (*Daniel & Danson, 2010*; *Radestock & Gohlke, 2011*). This flexibility-rigidity balance, governed by changes in protein dynamics, underscores the role of $\Delta C_p^{\ddagger}$ in optimizing catalytic activity under varying physiological conditions (*Arcus et al., 2022*).

While changes in $\Delta C_p^{\ddagger}$ are proposed to be very significant in explaining non-Arrhenius behavior in enzymatic reactions, alternative perspectives suggest that other mechanisms may be equally or more important. It has been proposed that protein dynamics and entropic contributions from the enzyme's environment play a significant role, emphasizing the importance of transition-state preorganization and intrinsic enzyme flexibility (*Lear et al., 2023*). Additionally, it has been highlighted how the ruggedness of an enzyme's energy landscape and its dynamic transitions could modulate reaction rates independently of $\Delta C_p^{\ddagger}$ (*Arcus et al., 2022*). These studies challenge the primacy of heat capacity changes

and propose that dynamic coupling and structural reorganizations are key contributors to the observed temperature dependencies.

In a complementary analysis, it was argued that the non-Arrhenius behavior of enzymatic reactions is better attributed to changes in the enzyme's configurational entropy rather than $\Delta C_p^{\ddagger}$ (*Warshel, Bora & Chu, 2021*). *Warshel, Bora & Chu (2021)* identified the interplay between protein motions and catalytic efficiency, proposing that deviations from the classical Arrhenius equation arise from entropic effects tied to conformational shifts in the enzyme-substrate complex. Instead, it likely results from a combination of factors, including energy landscape properties, dynamic coupling, and entropic contributions, reflecting the complexity of enzyme-catalyzed reactions.

In an attempt to simplify the analysis of a pseudo-peroxidase, this study chose MMRT to quantify the $\Delta C_p^{\ddagger}$ for myoglobin and compare that to a true peroxidase, lactoperoxidase, and to an inorganic catalyst, $Cu^{2+}$.

The application of $\Delta C_p^{\ddagger}$ analysis to systems like myoglobin may be considered to offer valuable insights into their pseudo-peroxidase activity. For example, molecular dynamics studies and crystallographic analyses have revealed distinct structural and dynamic differences that align with observed $\Delta C_p^{\ddagger}$ values (*Hobbs et al., 2013*; *Lear et al., 2023*). This approach provides a framework for characterizing enzymes that blur the lines between performing a naturally selected enzymatic reaction and an unintended catalyzed reaction.

To this end, the well-documented reaction of hydrogen peroxide receiving an electron from a common electron donor, 2,2′-azino-bis (3-ethylbenzothiazoline-6-sulfonic acid) (ABTS), was explored over available temperature windows for three different catalysts. The reaction was catalyzed in both pH 5 and pH 7 conditions by either the copper ion, the protein myoglobin, or the enzyme lactoperoxidase (5 being the pH at which the copper ion could catalyze the reaction to assure comparability, and 7 being more accurate to physiological conditions). The expectation was to see if there would be a difference in the temperature-dependence based on the mechanism of the catalysis for the three catalysts. If a molecule had a significant polypeptide chain contribution to the catalyzed reaction, then a plot of the $\ln(k_{cat})$ *vs.* temperature for that catalyst would have greater curvature, and its fit would reveal a larger $\Delta C_p^{\ddagger}$. However, should the catalyst not engage in a specific residue-dependent catalytic process, then the curve will collapse back into conventional kinetic models and reveal a low $\Delta C_p^{\ddagger}$. By this procedure, a gradient of catalytic contribution across the various catalysts can be theoretically obtained with the position of myoglobin in this range of catalysts, indicating the degree of its biocatalysis.

## MATERIALS AND METHODS

Equine heart myoglobin (MYO), Bovine Hemoglobin (HEMO), bovine milk lactoperoxidase (LP), 2,2′-azino-bis(3-ethylbenzothiazoline-6-sulfonic acid (ABTS)), 2-(N-morpholino)ethanesulfonic acid (MES), 4-(2-hydroxyethyl)-1-piperazineethanesulfonic acid (HEPES), $MgSO_4$, hydrogen peroxide, and $CuSO_4$ were purchased from Sigma Millipore and used without further purification.

Reaction rate data was collected, following the pattern in the literature (*Carlsen, Skovgaard & Skibsted, 2003*). The mechanism described in this reference was presumed to

be occurring in these reactions for myoglobin with possibly a similar mechanism for the other biocatalysts occurring for the other catalysts as well. To maintain a constant pH and ionic strength, MES and HEPES buffers were used to maintain a pH of 5 and 7, respectively, for all catalysts, and concentrations of $MgSO_4$ were used to match solutions' ionic strength. A solution at pH 5 was chosen because the copper ion is not an effective catalyst at a higher pH. Data was collected on the three biocatalysts at pH 7, despite a lack of inorganic catalyst data for comparison at this pH for insight into conditions similar to physiological conditions.

The color change of ABTS from its reduced to oxidized form was monitored at 730 nm to determine reaction rate. Spectral data was collected on a Vernier visible spectrometer by monitoring the initial change in absorption of ABTS at 730 nm as the compound released an electron to hydrogen peroxide, forming the ABTS radical cation (*Carlsen, Skovgaard & Skibsted, 2003*). The rate of this color change was directly related to the rates of the enzymatic catalysis. At this wavelength, an extinction coefficient for ABTS of 15,000 $M^{-1}cm^{-1}$ was used. MES buffer was used at pH 5 for HEMO, MYO, LP and $Cu^{2+}$. All biological catalysts were used in their ferric state as determined by the Soret band at 409 nm, in the case of MYO, for example. Additionally, the absorbance spectra between 450 and 650 nm was used to identify the oxidation state of the biocatalysts. In the presence of hydrogen peroxide, the spectra shifts for the proteins to a spectra that appears to be in the oxy from. Because it is well established that the ferrylmyoglobin species results in the presence of hydrogen peroxide it is most likely that this is the oxidation state of the biocatalysts when the ABTS donates an electron (*Barrick & Baldwin, 1989*; *Raven et al., 2007*). The spectra for these species are included in the Supplemental File associated with this work.

Experiment solutions contained 20 mM MES/HEPES buffer pH 5 or pH 7 and 1 mM ABTS. Reaction solutions that were catalyzed by MYO, LP and HEMO also had 13 mM $MgSO_4$. Hydrogen peroxide concentrations varied from 32 to 1,013 μM.

Reaction temperatures were controlled with both a Perkin Elmer Peltier temperature programmer 6 and a Thermo Fisher scientific hotplate. Temperature ranges for these experiments were as broad as the catalysts remained active. The ultimate range between the catalysts was 275 to 340 K. Enzyme denaturing and nonenzymatic peroxide degradation was checked by performing a control experiment in which sample temperatures were raised to maximum values and then lowered to room temperature before running samples and comparing these runs to samples that were run at room temperature. Temperatures that resulted in a statistical loss of enzymatic activity after raising to these temperatures were not used in data. This test was performed to capture any loss in activity due to catalyst deactivation or substrate concentration changes at elevated temperatures.

To give an effective comparison between the desired hydrogen peroxide decomposition reaction catalyzed enzymatically and "non enzymatically," a simple metal ion catalyst was found that would lower the activation energy of the reaction. A handful of transition metal ions were explored along with the iodide ion, which is renowned as a catalyst for the decomposition of hydrogen peroxide. The iodide ion was not chosen, however, because it

is also known that precipitates can form side reactions resulting in products that contain iodine derivatives. The copper (II) ion was found to be a stable catalyst at pH 5, although with markedly inefficient performance. This metal ion and the pH at which it catalyzes the peroxide reduction were used as a baseline to which the other catalysts were compared to determine the significance of their $\Delta C_p^\ddagger$ value, since the absence of macromolecular structures contributing to the reaction would ideally result in a measured $\Delta C_p^\ddagger$ of zero. Thus, the catalysis of the hydrogen peroxide degradation with ABTS as an electron donor serves as a control of an inorganic catalyst to compare to the biological catalysts.

All resulting fitting to either the Michaelis–Menten (MM) to obtain $V_{max}$ values or to the MMRT equation was done using Kaleidagraph and Excel. MM fitting was used to identify $V_{max}$ values. MMRT fitting was initially done to identify $\Delta H$ values at the apex of the curve. When there was no indication of an apex, fitting was not used to identify the apex. MMRT was then fit to identify the $\Delta C_p^\ddagger$.

The $\Delta C_p^\ddagger$ for an enzymatic reaction is a measurement of the change in the heat capacity of the reaction transition state compared to that of the bound catalyst and the substrate. Again, there is much discussion as to the contribution of this change, but it is arguably significantly related to changes in vibrational modes due to changes in the catalyst (*Hobbs et al., 2013*; *Arcus et al., 2016*, *Arcus & Pudney, 2015*). These changes in catalyst molecules can again occur in many ways with the most likely being due to adjustments in the polypeptide chain. It is following these lines of understanding that the $\Delta C_p^\ddagger$ of these different catalysts be considered a useful tool in the characterization of the whole catalyst's contribution to a catalyzed reaction as opposed to an unintended reaction in which relatively few atoms of the catalyst participate.

A follow-up comparison between two crystal structures of myoglobin from the literature at pH 5.2 and 6.8 were compared using a structure overlay software, SPDB viewer. The PDB files for these two structures can be found in the following reference (*Hersleth et al., 2007*). In these experiments, the heme molecules with their respective iron ions were forced to overlap, and relative distances of various corresponding atoms in the different polypeptide chains were compared to provide a rough volume difference of myoglobin at the different pH.

## RESULTS

$Cu^{2+}$ as a catalyst for the ABTS peroxide reaction was not found in the literature by this lab. However, there are references that suggest that coordinated $Cu^+$ can catalyze ABTS reduction in the presence of hydrogen peroxide (*Urbański & Beresewicz, 2000*; *Ma et al., 2020*). It is a possibility that the presence of the organic buffer may be involved in contributing to this reaction as well. This reaction mechanism may be further explored in the future. Various metal ions were experimented with to identify an effective non-enzymatic catalyst that was stable and consistent without any obvious sign of side reactions, with the catalyst ultimately selected being $Cu^{2+}$. The low efficiency of $Cu^{2+}$ as a catalyst required a high concentration of copper (II) sulfate in these experiments, resulting in an increase in conductivity. This high ionic strength was matched for all other biological catalysts using $MgSO_4$.
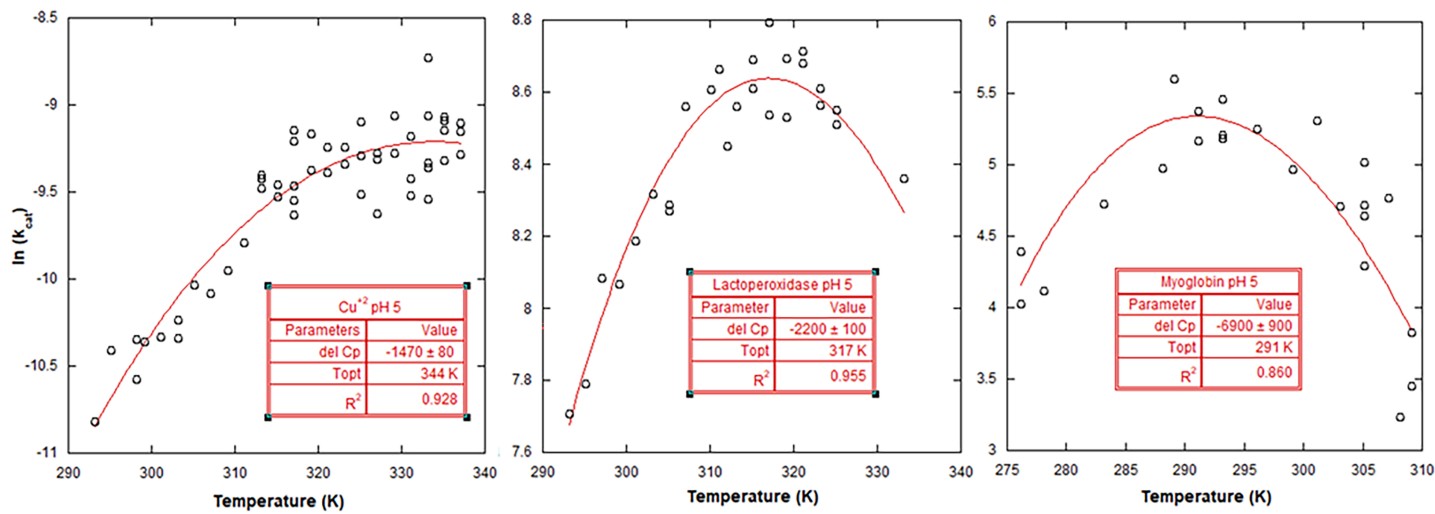

**Figure 2 MMRT fits to catalytic activity data at pH 5.** From left to right, the scatter plots and MMRT equation fits for: Cu$^{2+}$ (pH 5), lactoperoxidase (pH 5), and myoglobin (pH 5). The units for the $\Delta C_p^{\ddagger}$ are J mol$^{-1}$ K$^{-1}$.

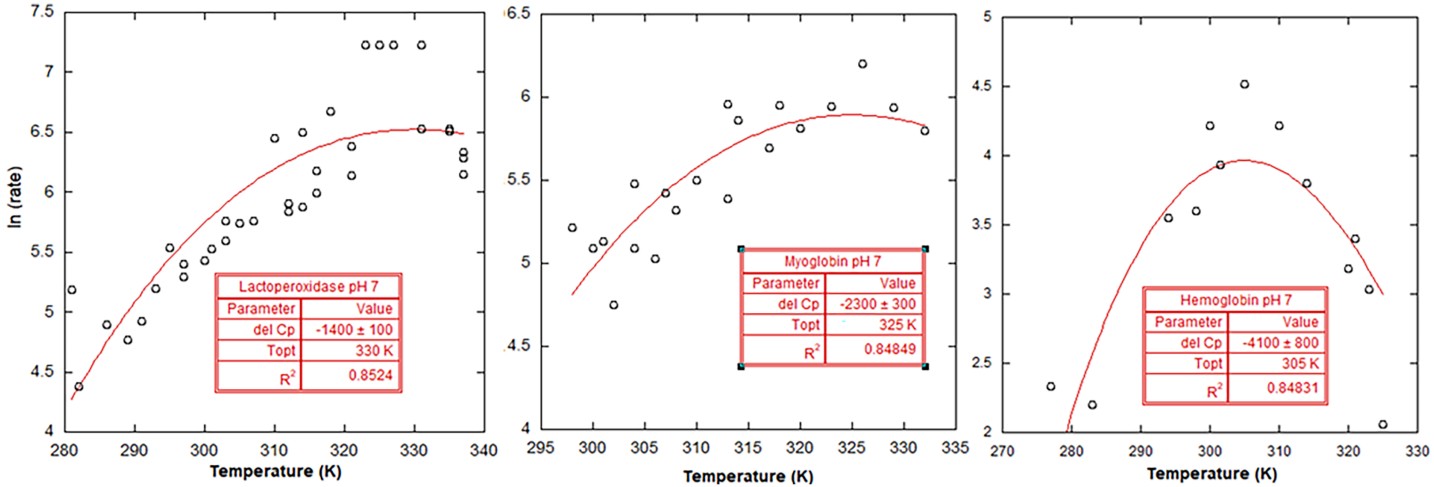

**Figure 3 MMRT fits to catalytic activity data at pH 7.** From left to right, the scatter plots and MMRT equation fits for: lactoperoxidase (pH 7), myoglobin (pH 7), and hemoglobin (pH 7). The units for the $\Delta C_p^{\ddagger}$ are J mol$^{-1}$ K$^{-1}$.

The resulting ln (k$_{cat}$) values at room temperature at a pH of 5 for the three different catalysts followed the expected reaction efficiency, with the relative values being largest for LP, followed by the MYO and then the Cu$^{2+}$ ion (7.79, 5.25, and −10.6), as shown in Fig. 2. The same expected trend was seen for LP and MYO at room temperature and pH 7 with ln (k$_{cat}$) values of 5.53 and 3.31, respectively, as shown in Fig. 3. Cu$^{2+}$ ion did not show any catalytic activity at pH 7.

Temperature ranges were chosen with an effort to identify non-Arrhenius behavior, without reaching temperatures that deactivated the biological catalysts. As mentioned above, this was checked by heating sample solutions containing catalysts, and then lowering the temperature of the sample solutions to verify that the heating did not result in

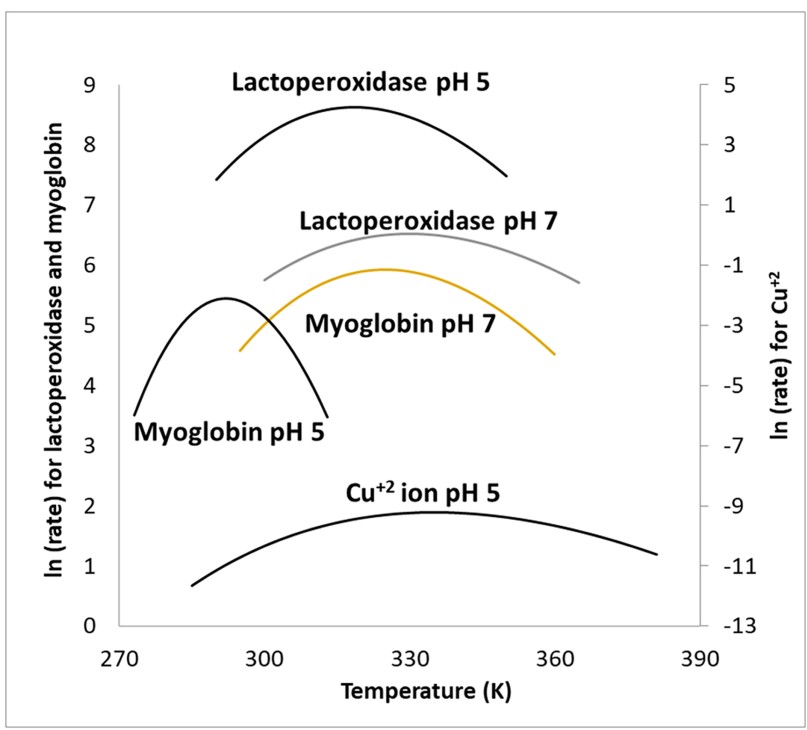

**Figure 4 Overlay of MMRT fits for various catalysts.** The resulting fits for the different catalysts at their different pH. A "tighter" curve denotes a higher $\Delta C_p^{\ddagger}$. Note that the copper ion has it's own ln (rate) scale on the right side of the graph.

a loss of enzyme activity once the temperature was lowered back down (the hydrogen peroxide was added last in these experiments). Additionally, higher temperatures were avoided when it was evident that the concentration of hydrogen peroxide began to deviate from the calculated added concentration (compared to reaction rates without heating), due to thermal decomposition of the peroxide. This was checked by heating sample solutions at higher temperatures and then lowering the temperature of the sample before the catalyst was added. For the temperature ranges chosen for the different catalysts there was not a significant decrease in reaction rate due to hydrogen peroxide concentration changes nor due to the catalyst, compromise. Rate values ($k_{cat}$) were calculated using the obtained $V_{max}$ values divided by the catalyst concentration. $V_{max}$ values were found by varying concentration of hydrogen peroxide and fitting the data to the Michaelis–Menten equation. The ln ($k_{cat}$) was plotted *vs.* the temperature of the reactions (Figs. 2 and 3). Figure 4 summarizes the fits for the different catalysts at their respective pH levels, showing the comparative curves and the difference between the $\Delta C_p^{\ddagger}$ values using a secondary y axis for the $Cu^{2+}$ ion.

A comparison of myoglobin structures from the literature showed that there is a shift in the distance between the iron atom and a collection of atoms in various the polypeptide residues with the pH 5.2 crystal structure being slightly more compact than the crystal structure at pH 6.8 (Fig. 5).

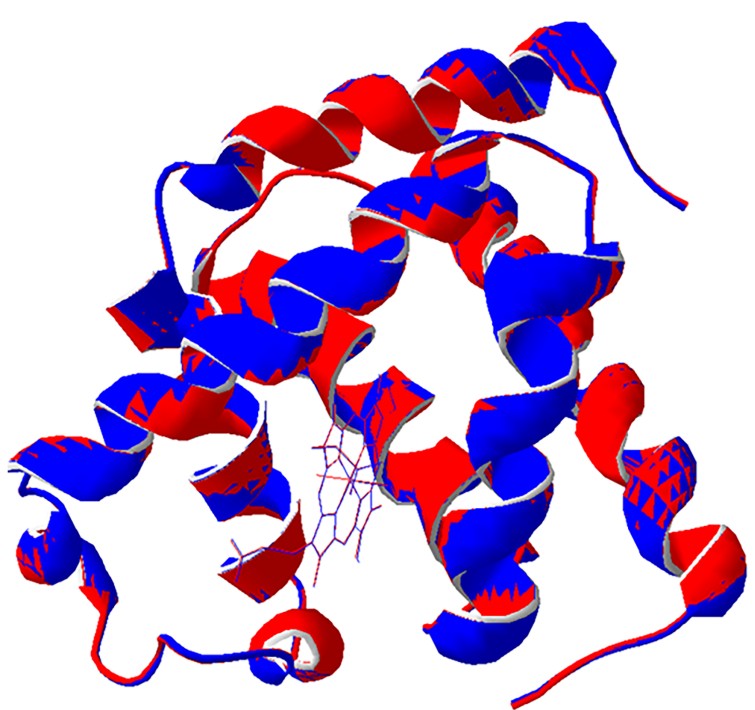

| Distance from residue to Fe atom | | | |
|---|---|---|---|
| Residue | Myo 5.2 | Myo 6.8 | Difference in Distance |
| OXT153 | 14.18 | 13.98 | -0.2 |
| ALA19 | 21.46 | 21.47 | 0.01 |
| GLY1 | 24.7 | 24.48 | -0.22 |
| LYS56 | 18.84 | 18.81 | -0.03 |
| LYS79 | 21.4 | 21.44 | 0.04 |
| LYS96 | 10.14 | 10.22 | 0.08 |
| GLY121 | 29.82 | 29.93 | 0.11 |
| GLU41 | 12.13 | 12.15 | 0.02 |
| LYS45 | 13.14 | 13.09 | -0.05 |
| LYS50 | 20.89 | 21 | 0.11 |
| SER3 | 26.27 | 26.28 | 0.01 |
| GLY124 | 26.55 | 26.62 | 0.07 |
| PRO120 | 28.49 | 28.61 | 0.12 |
| GLU148 | 19.94 | 19.89 | -0.05 |
| VAL10 | 22.31 | 22.38 | 0.07 |
| ALA144 | 17.55 | 17.49 | -0.06 |
| GLY80 | 29.04 | 29.52 | 0.48 |
| ALA19 | 20.23 | 20.33 | 0.1 |
| MET55 | 16.19 | 16.18 | -0.01 |
| GLY150 | 18.48 | 18.45 | -0.03 |
| | | Sum | 0.57 |
| | | Average | 0.0285 |

**Figure 5 Overlay of Myoglobin at different pH showing a slight expansion of the polypeptide chain.** Overlay of myoglobin molecules at pH 5.2 (red) and 6.8 (blue) (*Hersleth et al., 2007*). The blue predominates on the outer edges of the protein (portions facing the heme) while the red predominates on the inner edges (portions away from the heme) suggesting a conformational change that tightens the molecule at a lower pH. The table shows measurements from specific residues to the Heme Fe atom. This is consistent with the findings that at pH 7, myoglobin, the form that is more relaxed in its structure, has a smaller $\Delta C_p^\ddagger$.

## DISCUSSION

The $\Delta C_p^\ddagger$ for a catalyst has been suggested to be related to a change in the low-frequency vibrational modes of the catalyst's polypeptide structure (*Arcus & Pudney, 2015*). It has been suggested that this change is associated with the transition state complex of the catalyst mid-reaction possessing a structure that is substantially different from the ground state of the catalyst as illustrated in Fig. 1. The underlying assumption of this connection in terms of its relevancy to data analysis is that the curve demonstrated in the ln ($k_{cat}$) *vs* temperature plot is defined by the MMRT equation and its terms, rather than an alternative process that results in decreasing enzymatic activity as temperature increases in the absence of catalyst/substrate denaturation. Should this assumption be false, then the efficacy of these findings would be subject to greater scrutiny.

Discussing the calculated $\Delta C_p^\ddagger$ for the $Cu^{2+}$ ion, it was hypothesized that this catalyst would have a $\Delta C_p^\ddagger$ closest to zero and thus being almost perfectly Arrhenius in its behavior. There could be arguments based on a theoretical complex between the participants, that the size of a complex between the ABTS molecule, the hydrogen peroxide, and the $Cu^{2+}$ ion, and perhaps an included layer or two of solvating water molecules, that a non zero $\Delta C_p^\ddagger$ could result. It could also be that the fitting in this case was forced and that there is actually more Arrhenius behavior that perceived.

One of the variables in the MMRT equation that is not inherently in the Eyring equation is the optimal temperature ($T_{opt}$) for the reaction. In this works the temperature at the peak of the MMRT fit was chosen as the $T_{opt}$. This need for the $T_{opt}$ occurs because as the MMRT equation was derived, it is seen (Eqs. (3) and (4)) the $\Delta C_p^\ddagger$ is a related to the difference in the temperature at which the reaction occurs and the $T_{opt}$. Indeed, it is the experience of this work that choosing different $T_{opt}$ values significantly influences the resulting $\Delta C_p^\ddagger$. The need for this reference temperature is an artifact of expanding the Eyring equation into the MMRT and can create a bit of a problem when analyzing the data using non-linear regression. It may be the case that when fitting the MMRT to data, the need to provide an initial value for the $T_{opt}$ results in a MMRT fit that would not typically exist because the reaction does not actually have a $T_{opt}$. A reaction would not have a $T_{opt}$, for example, if it followed strict Arrhenius kinetics. This should be accounted for when considering the meaning of the $\Delta C_p^\ddagger$ for the $Cu^{2+}$ catalyzed reaction. Catalysis with the $Cu^{2+}$ would not be expected to have a $T_{opt}$, therefore, the resulting fit that suggests a $\Delta C_p^\ddagger$ may be erroneous because a $T_{opt}$ is placed into the fitting process to see how well it fits the MMRT equation. In other words, because the $T_{opt}$ is a required variable to fit the MMRT, an initial inspection of the data to find a potential $T_{opt}$ is required. This does not mean that an MMRT fit to the $Cu^{2+}$ data is useless at contributing to the understanding of the nature of catalysis of the biological catalysts. The data for the $Cu^{2+}$ serves as a comparison and, in a way, a control for the hydrogen peroxide degradation reaction. The $Cu^{2+}$ is catalyzing the degradation of hydrogen peroxide using ABTS as an electron source, but the exact mechanism is not known. However, the similarity for these reactions, the common chemicals, and the possible common side reactions, should represent a good approximation, if not a perfect comparison, between the inorganic and the organic catalysts.

A significant portion of the data analysis is determining the $T_{opt}$ for the given catalyst. Once the $T_{opt}$ has been identified, then the symmetry of the MMRT is enough to result in a fit that estimates the $\Delta C_p^\ddagger$. In the case that a $T_{opt}$ is incorrectly identified on a reaction that does not have a $T_{opt}$, then the data should be close to linear, and the resulting $\Delta C_p^\ddagger$ could be a representation of the minimal $\Delta C_p^\ddagger$ for a given reaction. Thus, the $\Delta C_p^\ddagger$ calculated here from fitting the data for the $Cu^{2+}$ is quite likely not an actual $\Delta C_p^\ddagger$ for this reaction as, again, for an inorganic catalyst there would not be expected to be a significant $\Delta C_p^\ddagger$. However, because it could be argued that there may actually be a $\Delta C_p^\ddagger$ for reactions catalyzed by an inorganic catalyst, we must treat the data the same way in trying to find a $T_{opt}$. An example of a way that such a reaction might have a $\Delta C_p^\ddagger$ for such a small catalyst might be from a change in hydration spheres around charged species in the reaction transition state.

Thus, if the $\Delta C_p^\ddagger$ for the inorganic catalyzed reaction is real, then the value is meaningful as a comparison for this specific reaction. If it is not meaningful and a $T_{opt}$ was incorrectly fit to the data, the $\Delta C_p^\ddagger$ represents the potential error in data analysis when comparing catalysts for this reaction. In all MMRT fits for this research, the $T_{opt}$ was chosen to minimize error, maximizing the $R^2$ value.

In both cases, either the $Cu^{2+}$ ion having a $\Delta C_p^\ddagger$ or the fitting actually being made to linear data, the resulting information is helpful in showing the extent to which MYO, as a

catalyst for this reaction, has a significantly larger $\Delta C_p^{\ddagger}$ when compared with the $Cu^{2+}$. The additional comparison between LP and MYO further shows the difference in the $\Delta C_p^{\ddagger}$ of a peroxidase and a pseudo-peroxidase. The relatively large $\Delta C_p^{\ddagger}$ for MYO compared to LP at pH 5 would suggest that MYO has more of a transition-state shift in its reaction.

HEMO was also included as a catalyst at pH 7 since it shares some similar reaction site characteristics with MYO but with a tetrameric structure as opposed to a monomeric structure. This difference in quaternary structure between the two catalysts can be observed in their kinetic behavior at the same pH with HEMO demonstrating a markedly higher $\Delta C_p^{\ddagger}$ at pH 7. This suggests that HEMO's structure undergoes a greater reduction in low frequency vibrational modes when in a catalyst-transition state complex as compared to MYO, a property which may also be related to the cooperativity of HEMO's multiple binding sites.

A follow up experiment to look at any conformations changes in the literature of myoglobin at different pH yielded confirming data. Figure 5 shows the overlay of myoglobin molecules at different pH. The decrease in the $\Delta C_p^{\ddagger}$ at the higher pH is consistent with the more relaxed conformation and its impact on catalytic efficiency (*Hersleth et al., 2007*). The literature shows that as biocatalyst structures relax due to unfavorable catalytic mutations, the $\Delta C_p^{\ddagger}$ also decreases (*Hobbs et al., 2013*).

When comparing these two catalysts, it should also be mentioned that the mechanisms for the reactions are surely at least slightly different and may be significantly different (*Cupp-Sutton & Ashby, 2021*). The studies that looked at the reaction mechanisms for these catalysts do not directly compare the reaction mechanisms with ABTS as an electron donor.

Another discussion should be had about the relative difference between the $\Delta C_p^{\ddagger}$ for the pseudo-peroxidases and the peroxidase. Large negative $\Delta C_p^{\ddagger}$ (between $-6$ and $-10$ kJ mol$^{-1}$ K$^{-1}$) do not seem to be unusual in the literature (*Hobbs et al., 2013*; *Arcus et al., 2016*). This difference may hold in it something that may be a good distinguisher between a peroxidase and a pseudo-peroxidase but there is nothing that stands out at this time. It would seem intuitive that if a larger $\Delta C_p^{\ddagger}$ relates to a larger polypeptide contribution then either this approach to distinguish between a peroxidase and a pseudo-peroxidase is not effective or that myoglobin might be better considered as a peroxidase.

## CONCLUSIONS

The resulting data, when fit to the MMRT equation, suggest that all of the catalysts show signs of a $\Delta C_p^{\ddagger}$. As expected, the $\Delta C_p^{\ddagger}$ for the inorganic catalyst ($Cu^{2+}$) is the smallest. Unexpectedly, the LP had a lower $\Delta C_p^{\ddagger}$ than MYO at pH 5. In addition to these findings, MYO demonstrated significantly altered $\Delta C_p^{\ddagger}$ when the reaction occurred at pH 7. This alteration (*i.e.*, "flattening") to the curve is largely consistent with the change observed in LPO when the reaction was performed at pH 7, indicating a degree of pH dependence in the heat capacity shift of individual catalysts. It is possible that at the higher pH the myoglobin has a more relaxed structure that would generally be related to to a smaller $\Delta C_p^{\ddagger}$. Figure 5 reinforces the conclusions drawn from the structural analysis, showing the pH-dependent conformational changes in myoglobin. These findings are consistent with

studies that demonstrate significant changes in heat capacity for myoglobin under varying pH conditions, particularly in the range of acidic to neutral pH (*e.g.*, pH 5 to pH 7). This variation is attributed to alterations in the hydration shell, increased flexibility of the protein backbone, and changes in structural stability that occur as the protonation states of key residues shift, impacting the overall conformational dynamics and thermodynamic properties of the enzyme (*Zhu & Brewer, 2002*).

## ACKNOWLEDGEMENTS

This article is dedicated to the memory of Grant Barron (2000–2024), a dedicated colleague and friend whose tireless efforts made this research possible. This manuscript was developed with the assistance of ChatGPT, an AI language model. The AI was used to identify relevant research materials, suggest related studies, and assist in formatting references according to the required journal style. Additionally, it helped refine the clarity and structure of the text, but all final content was reviewed and approved by the authors.

### Funding

This work was supported by the College of Science at Utah Valley University. The funders had no role in study design, data collection and analysis, decision to publish, or preparation of the manuscript.

### Grant Disclosures

The following grant information was disclosed by the authors:
College of Science at Utah Valley University.

### Competing Interests

The authors declare that they have no competing interests.

### Author Contributions

- Collin Tuttle performed the experiments, prepared figures and/or tables, authored or reviewed drafts of the article, and approved the final draft.
- Michael Hannesson performed the experiments, prepared figures and/or tables, and approved the final draft.
- Amy Henrichsen performed the experiments, authored or reviewed drafts of the article, and approved the final draft.
- Lily Hainsworth performed the experiments, authored or reviewed drafts of the article, and approved the final draft.
- Camille Condie performed the experiments, authored or reviewed drafts of the article, and approved the final draft.
- Aj Whitesides performed the experiments, authored or reviewed drafts of the article, and approved the final draft.
- Archel Oren performed the experiments, authored or reviewed drafts of the article, and approved the final draft.

- Simeon Tanner performed the experiments, authored or reviewed drafts of the article, and approved the final draft.
- Benjamin Terry performed the experiments, authored or reviewed drafts of the article, and approved the final draft.
- Jacob Cannon performed the experiments, authored or reviewed drafts of the article, and approved the final draft.
- Jeremy Johansen performed the experiments, authored or reviewed drafts of the article, and approved the final draft.
- Alisha Bhatia analyzed the data, prepared figures and/or tables, authored or reviewed drafts of the article, and approved the final draft.
- Daniel Scott conceived and designed the experiments, prepared figures and/or tables, authored or reviewed drafts of the article, and approved the final draft.

## Data Availability

The raw measurements are available in the Supplemental File.

## Supplemental Information

Supplemental information for this article can be found online at http://dx.doi.org/10.7717/peerj.19205#supplemental-information.

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
