# Peer review of "A case for myoglobin-macromolecular rate theory applied to pseudo peroxidase kinetics"

_PeerJ, doi:10.7717/peerj.19205_

## Round 0.1 · original submission · Major Revisions

· Academic Editor

Major Revisions

Kindly revise your submission by addressing the reviewers' comments and include a rebuttal letter. We appreciate your consideration of submitting your work to PeerJ.

·

Basic reporting

The manuscript describes the characterization of pseudo peroxidase activity of myoglobin at different pH´s and temperatures and its comparison with other systems (copper and other hemeproteins). Moreover, the authors aim to explain the participation of polypeptide chains in this process.

It is important to mention that the reaction in question was studied previously, and the present manuscript is not used for the result description. In addition, the results are limited, and the contribution to the science is not clear. Moreover, the authors aim to explain the participation of polypeptide chains in this reaction but they did not present the evidence necessary to describe its participation.

Experimental design

The methodology section of the present manuscript has limitations and promotes poor results validation. The following are mentioned specific aspects:

- The temperature range used is not mentioned (Lines 191-192).
- What is the objective for using copper as an experimental treatment? Is it a control of this reaction?
- Line 186 730 nm?? What is the fundament to use this wavelength? Because it was not another? Is wavelength to monitor the concentration of all proteins or hydrogen peroxide decomposition?
- The method of monitoring the peroxidase activity is not clear. It aspect confers a confusing interpretation of results.
- Is the oxidative state of Mb important for replicating the peroxide activity? The state oxidation is fundamental to the reactivity of Mb, exist different examples of this aspect in the literature.
-The method section does not present the literature necessary to support its use.
-it is not clear how to obtain experimentally the evidence for confirm the participation of polypeptide chains in this reaction.

Validity of the findings

In general, the findings contemplated by authors are necessary to use different methods that are not employed. The principal's comments to respect are described:

-The sentence “Copper (II) ion (Cu2+) as a catalyst for the ABTS peroxide reaction was not found in the literature by this lab” is confusing. In this sense, the Cu ions in the ABTS-H2O2 system are used to determine glucose detection. doi:10.1088/1742-6596/1676/1/012126.
- The methodology for obtaining the structural differences between the myoglobin structures to different pH´s described in Fig. 5, is not clear. However, an in silico strategy for determining these structural changes is for molecular dynamic simulation. Moreover, this is a modular part of this research, and the authors do not present evidence for this.
-The discussion of results is limited, and the apport science is not clear. Moreover, the literature on the characterization of myoglobin peroxidase activity is diverse.
-It is necessary to mention the state of myoglobin oxidation during the reaction. This is fundamental to better describe the mechanism involved. In this sense, metmyoglobin is well described as the reaction with hydrogen peroxide, with ferrylmyoglobin as the reaction product.
-It is recommended to focus on the characterization of spectral changes, principally for the Soret band from myoglobin, to obtain major structural evidence during the reaction, especially about the heme cavity.

·

Basic reporting

The manuscript is clear and well-prepared. The historical context is presented effectively, and the results align with the stated hypothesis. The literature is thoroughly analyzed and effectively utilized. Additionally, the ideas presented, particularly the generalization to other enzymatic systems, will be well received by the specialized community.

Experimental design

The experimental design and detailed explanations of the materials and methods are highly appreciated. The paper is quite engaging and could serve as a valuable textbook. Therefore, I believe it will be well-received by both specialists and postgraduate students.

Validity of the findings

Although the concept presented is not new, the design and historical context effectively engage readers. One minor point to consider relates to Figure 4; the authors should clarify whether the macromolecular models originate from a PDB deposit or are computational models. In any case, it is essential for the authors to provide a detailed description of their original work and include a brief comment on how the origin of these coordinates relates to their findings.

Additional comments

This is a very interesting and well-presented manuscript. I would like to address the origin of the models presented in Figure 4. Are they sourced from PDB deposits, or were they derived from computational calculations? Regardless of their origin, please include this information in the manuscript and add a comment on the importance of their methodological origin with respect to the findings discussed.

Reviewer 3 ·

Basic reporting

.

Experimental design

.

Validity of the findings

.

Additional comments

Attached.

Annotated reviews are not available for download in order to protect the identity of reviewers who chose to remain anonymous.

---

## Round 0.2 · Minor Revisions

· Academic Editor

Minor Revisions

Please consider all reviewers' suggestions when submitting a revised version.

·

Basic reporting

The new version of the present manuscript is more comprehensible; however, it is recommended that the introduction section be adjusted. Only present the information necessary to express the background and objectives.

Experimental design

The methods section is major presented. But the pdb code of each protein structure mentioned is suggested. Additionally, the sentence “In the presence of hydrogen peroxide, the oxy from of the proteins were evident” promotes a confuse. The interaction between Mb and hydrogen peroxide forms a hyper-oxidative state of Mb named ferrylmyoglobin, and not OxyMb. Ferrylmyoglobin and OxyMb have a very similar visible spectrum in the Q region. How determine which oxidative form is present in your experiments?

Validity of the findings

No comment.

Additional comments

It recommended a major revision to text aspects as spaces and commas.

Reviewer 3 ·

Basic reporting

No comment

Experimental design

No comment

Validity of the findings

No comment

---

## Round 0.3 · accepted · Accept

· Academic Editor

Accept

Thank you for addressing all reviewers' concerns. Your manuscript is now accepted in PeerJ!